# Common Fixed-Point and Fixed-Circle Results for a Class of Discontinuous *F*-Contractive Mappings †

**Pradip Debnath** 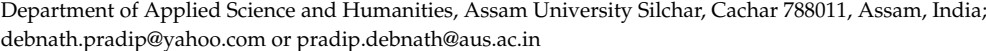

Department of Applied Science and Humanities, Assam University Silchar, Cachar 788011, Assam, India; debnath.pradip@yahoo.com or pradip.debnath@aus.ac.in

† The author dedicates this paper to the wonderful mathematician Prof. Billy E. Rhoades for his outstanding works in Fixed Point Theory which have provided the younger mathematicians enough to work for generations.

**Abstract:** The exploration of contractive inequalities which do not imply the continuity of the mapping at fixed points was an interesting open problem for quite some time. A significant amount of progress was made in the last two decades towards the solution of this problem. In the current paper, we attempt to address the question of discontinuity at fixed point with the help of *F*-contractions in a metric space. We establish a common fixed-point (CFP) result for such contractive mappings and investigate its discontinuity at the CFP. A fixed-circle result is also obtained consequently.

**Keywords:** discontinuity; fixed point; common fixed point; *F*-contraction; complete metric space; fixed-circle

**MSC:** 47H10; 54H25; 54E50

## 1. Introduction and Preliminaries

The well-known contractive inequality due to Stefan Banach forces the mapping to be continuous. However, the investigation of discontinuity at fixed points was initiated by Kannan in his 1968–69 papers [1,2]. All the familiar mappings at that point of time exhibited continuity at the fixed points in spite of the fact that they admitted points of discontinuity within their domains [3–5]. In 1977, Rhoades [6] presented a comparative study of 250 contractive definitions and noticed that many of those definitions did not imply the continuity of the mappings in their respective domains. Motivated by this, in 1988, Rhoades [7] posed the open problem of exploring contractive definitions which do not compel the mapping to be continuous at fixed points. Pant [8], in 1999, gave a positive answer to this exploration by constructing mappings which exhibited discontinuity at their fixed points.

Fixed-point results for mappings with discontinuity are well sought-after because of their wide variety of applications in neural networks, character recognition, and the solution of non-negative sparse approximation problems [9–13]. Recently, this study gained immense popularity and various authors have provided affirmative solution to the problem posed by Rhoades (see Bisht and Rakocević [14], Pant et al. [15], Tas and Ozgur [16], Ozgur and Tas [17]).

In this paper, we attempt to address the issue of discontinuity at fixed point with the help of a class of *F*-contractive mappings in a metric space (MS). Throughout this manuscript, we use the standard symbol $\implies$ to denote "implies".

In 2012, Wardowski [18] defined the concept of *F*-contraction as follows.

**Definition 1.** *Let $F : (0, +\infty) \to (-\infty, +\infty)$ be a function which satisfies the following:*

**(F1)** *F is strictly increasing;*

**(F2)** *For each sequence $\{u_n\}_{n\in\mathbb{N}} \subset (0, +\infty)$, $\lim_{n\to+\infty} u_n = 0$ if and only if $\lim_{n\to+\infty} F(u_n) = -\infty$;*

**(F3)** *There is $t \in (0,1)$ such that $\lim_{u \to 0^+} u^t F(u) = 0$.*

Let $\mathcal{F}$ denote the class of all such functions $F$. If $(\mathcal{W}, \eta)$ is an MS, then a self-map $\Phi : \mathcal{W} \to \mathcal{W}$ is said to be an F-contraction if there exist $\delta > 0$, $F \in \mathcal{F}$, such that for all $\theta, \xi \in \mathcal{W}$,

$$\eta(\Phi\theta, \Phi\xi) > 0 \Rightarrow \delta + F(\eta(\Phi\theta, \Phi\xi)) \leq F(\eta(\theta, \xi)).$$

We divide the main results of this paper into two sections. The first section deals with a CFP result where the mappings under consideration are discontinuous at the CFP. In the second section, we present a fixed-circle result without assuming completeness of the MS. For some more interesting relevant works, we refer to [19–24].

## 2. Common Fixed Point with Discontinuity of the Contraction

In this section, we establish a CFP result and study discontinuity at the CFP. The following notation will be used.

$$\Lambda_0(\theta, \xi) = \max\{\eta(\theta, \xi), \eta(\theta, \Phi\theta), \eta(\xi, \Psi\xi), [\frac{\eta(\theta, \Psi\xi) + \eta(\xi, \Phi\theta)}{\eta(\theta, \Phi\theta) + \eta(\xi, \Psi\xi) + 1}]\eta(\theta, \xi)\}.$$

**Theorem 1.** *Let $(\mathcal{W}, \eta)$ be a complete MS and $\Phi, \Psi : \mathcal{W} \to \mathcal{W}$ be a pair of self-maps such that there exist $\delta > 0$ and $F \in \mathcal{F}$ satisfying*

*(i)* $\delta + F(\eta(\Phi\theta, \Psi\xi)) \leq F(\Gamma(\Lambda_0(\theta, \xi)))$ *for all $\theta, \xi \in \mathcal{W}$, where $\Gamma : \mathbb{R}^+ \to \mathbb{R}^+$ has the property $\Gamma(s) < s$ for each $s > 0$;*

*(ii)* *For a given $\epsilon > 0$, there exist $\kappa > 0$ such that $\epsilon < \Lambda_0(\theta, \xi) < \epsilon + \kappa$ implies that $\eta(\Phi\theta, \Psi\xi) \leq \epsilon$.*

*Then, the pair $\Phi, \Psi$ possesses a CFP, say $\omega$, and $\lim_{n \to +\infty} \Phi^n \theta \to \omega$, $\lim_{n \to +\infty} \Psi^n \theta \to \omega$ for each $\theta \in \mathcal{W}$. Moreover, $\Phi$ and $\Psi$ happen to be discontinuous at $\omega$ if and only if $\lim_{\theta \to \omega} \Lambda_0(\theta, \omega) \neq 0$ or $\lim_{\xi \to \omega} \Lambda_0(\omega, \xi) \neq 0$.*

**Proof.** Fix $\theta_0 \in \mathcal{W}$ such that $\theta_0 \neq \Phi\theta_0$ and $\theta_0 \neq \Psi\theta_0$. Construct the sequence $\{\theta_n\}$ by $\theta_{2n+1} = \Phi^{2n}\theta_0 = \Phi\theta_{2n}$ and $\theta_{2n+2} = \Psi^{2n+1}\theta_0 = \Psi\theta_{2n+1}$ for $n = 0, 1, 2, \ldots$. We denote $d_n = \eta(\theta_n, \theta_{n+1})$.

Using $(i)$ of the hypothesis, we have

$$
\begin{aligned}
F(\eta(\Phi\theta_0, \Psi\theta_1)) &\leq F(\Gamma(\Lambda_0(\theta_0, \theta_1))) - \delta \\
&< F(\Gamma(\Lambda_0(\theta_0, \theta_1))) \\
\Longrightarrow \eta(\Phi\theta_0, \Psi\theta_1) &< \Gamma(\Lambda_0(\theta_0, \theta_1)).
\end{aligned}
\tag{1}
$$

Now,

$$
\begin{aligned}
d_1 = \eta(\theta_1, \theta_2) &= \eta(\Phi\theta_0, \Psi\theta_1) \\
&< \Gamma(\Lambda_0(\theta_0, \theta_1)) \\
&= \Gamma(\max\{\eta(\theta_0, \theta_1), \eta(\theta_0, \Phi\theta_0), \eta(\theta_1, \Psi\theta_1), [\frac{\eta(\theta_0, \Psi\theta_1) + \eta(\theta_1, \Phi\theta_0)}{\eta(\theta_0, \Phi\theta_0) + \eta(\theta_1, \Psi\theta_1) + 1}]\eta(\theta_0, \theta_1)\}) \\
&= \Gamma(\max\{\eta(\theta_0, \theta_1), \eta(\theta_0, \theta_1), \eta(\theta_1, \theta_2), [\frac{\eta(\theta_0, \theta_2) + \eta(\theta_1, \theta_1)}{\eta(\theta_0, \theta_1) + \eta(\theta_1, \theta_2) + 1}]\eta(\theta_0, \theta_1)\}) \\
&\leq \Gamma(\max\{\eta(\theta_0, \theta_1), \eta(\theta_1, \theta_2), [\frac{\eta(\theta_0, \theta_1) + \eta(\theta_1, \theta_2)}{\eta(\theta_0, \theta_1) + \eta(\theta_1, \theta_2) + 1}]\eta(\theta_0, \theta_1)\}) \\
&\leq \Gamma(\max\{\eta(\theta_0, \theta_1), \eta(\theta_1, \theta_2)\}).
\end{aligned}
\tag{2}
$$

If $\eta(\theta_0, \theta_1) \leq \eta(\theta_1, \theta_2)$, then using condition $(ii)$ of the hypothesis and a property of $\Gamma$, we have that $\eta(\theta_1, \theta_2) < \Gamma(\eta(\theta_1, \theta_2)) < \eta(\theta_1, \theta_2)$, which is a contradiction. Thus, we must have $\eta(\theta_1, \theta_2) < \eta(\theta_0, \theta_1)$.

So, from (2),

$$d_1 = \eta(\theta_1, \theta_2) < \Gamma(\eta(\theta_0, \theta_1)) = \Gamma(d_0) < d_0.$$

Similarly, one can show that

$$d_2 = \eta(\theta_2, \theta_3) < \Gamma(\eta(\theta_1, \theta_2)) = \Gamma(d_1) < d_1.$$

Using mathematical induction, we obtain that

$$d_n = \eta(\theta_n, \theta_{n+1}) < \Gamma(\eta(\theta_{n-1}, \theta_n)) = \Gamma(d_{n-1}) < d_{n-1} \text{ for } n = 0, 1, 2, \ldots.$$

Thus, $\{d_n\}$ is a strictly decreasing sequence of positive reals and hence converges to, say $d$. If possible, suppose that $d > 0$. Obviously, there exists $p \in \mathbb{N}$ such that for $n \geq p$, we have

$$d < d_n < d + \kappa. \tag{3}$$

Using $(ii)$ of the hypothesis and the fact that $d_n < d_{n-1}$, we have that $d_n \leq d$ for all $n \geq p$, which contradicts (3). Thus, we have $d = 0$.

Next, using a similar technique as in [17] we can show that $\{\theta_n\}$ is Cauchy.

Since $\mathcal{W}$ is complete, there exists a point $\omega \in \mathcal{W}$ such that $\theta_n \to \omega$ as $n \to +\infty$. In addition, we have that $\Phi^n \theta_n \to \omega$ and $\Psi^n \theta_n \to \omega$ as $n \to +\infty$.

Next, we show that $\omega$ is a CFP of $\Phi$ and $\Psi$.

If possible, suppose that $\omega \neq \Psi\omega$. From condition $(i)$ of the hypothesis and generalizing inequality (1), we have that

$$
\begin{aligned}
\eta(\omega, \Psi\omega) &< \eta(\omega, \theta_{2n+1}) + \eta(\theta_{2n+1}, \Psi\omega) \\
&= \eta(\omega, \theta_{2n+1}) + \eta(\Phi\theta_{2n}, \Psi\omega) \\
&\leq \eta(\omega, \theta_{2n+1}) + \Gamma(\Lambda_0(\theta_{2n}, \omega)) \\
&= \eta(\omega, \theta_{2n+1}) + \Gamma(\max\{\eta(\theta_{2n}, \omega), \eta(\theta_{2n}, \Phi\theta_{2n}), \eta(\omega, \Psi\omega), \\
&\quad [\frac{\eta(\theta_{2n}, \Psi\omega) + \eta(\omega, \Phi\theta_{2n})}{\eta(\theta_{2n}, \Phi\theta_{2n}) + \eta(\omega, \Psi\omega) + 1}]\eta(\theta_{2n}, \omega)\}).
\end{aligned}
$$

Letting $n \to +\infty$ in the last inequality, we have that

$$\eta(\omega, \Psi\omega) \leq \Gamma(\eta(\omega, \Psi\omega)) < \eta(\omega, \Psi\omega),$$

which is a contradiction. Hence, $\omega = \Psi\omega$. Similarly, it follows that $\omega = \Phi\omega$. Thus, $\omega$ is a CFP of $\Phi$ and $\Psi$.

To prove the next part, let $\lim_{\theta \to \omega} \Lambda_0(\theta, \omega) = 0$ and $\lim_{\xi \to \omega} \Lambda_0(\omega, \xi) = 0$.

Using definition of $\Lambda_0(\theta, \omega)$, we have that

$$\lim_{\theta \to \omega}[\max\{\eta(\theta, \omega), \eta(\theta, \Phi\theta), \eta(\omega, \Psi\omega), [\frac{\eta(\theta, \Psi\omega) + \eta(\omega, \Phi\theta)}{\eta(\theta, \Phi\theta) + \eta(\omega, \Psi\omega) + 1}]\eta(\theta, \omega)\}] = 0$$

$$\implies \lim_{\theta \to \omega} \eta(\theta, \Phi\theta) = 0$$

$$\implies \Phi \text{ is continuous at } \omega.$$

Again, using definition of $\Lambda_0(\omega, \xi)$, we have that

$$\lim_{\xi \to \omega}[\max\{\eta(\omega, \xi), \eta(\omega, \Phi\omega), \eta(\xi, \Psi\xi), [\frac{\eta(\omega, \Psi\xi) + \eta(\xi, \Phi\omega)}{\eta(\omega, \Phi\omega) + \eta(\xi, \Psi\xi) + 1}]\eta(\omega, \xi)\}] = 0$$

$$\implies \lim_{\xi \to \omega} \eta(\xi, \Psi\xi) = 0$$

$$\implies \Psi \text{ is continuous at } \omega.$$

The converse of this part can also be proved using similar techniques.

Hence, at least one $\Phi$ and $\Psi$ is discontinuous at $\omega$ if and only if $\lim_{\theta \to \omega} \Lambda_0(\theta, \omega) \neq 0$ or $\lim_{\xi \to \omega} \Lambda_0(\omega, \xi) \neq 0$. $\quad \square$

**Remark 1.** *If $\Phi = \Psi$ in the above theorem, we obtain a fixed point result.*

Below, we provide an example to validate Theorem 1.

**Example 1.** *Let $\mathcal{W} = [0,4]$ be endowed with the usual metric $\eta$. Define $\Phi, \Psi : \mathcal{W} \to \mathcal{W}$ by*

$$\Phi\theta = \begin{cases} 2, & \text{if } \theta \in [0,2] \\ 0.85, & \text{if } \theta \in (2,4], \end{cases}$$

*and*

$$\Psi\theta = \begin{cases} 2, & \text{if } \theta \in [0,2] \\ 0.88, & \text{if } \theta \in (2,4]. \end{cases}$$

*Then, $\omega = 2$ is a CFP of $\Phi, \Psi$ and both the mappings are discontinuous at $\omega = 2$ (see Figure 1).*

*$\Phi$ and $\Psi$ satisfy condition $(i)$ of Theorem 1 with $\delta = \ln 2$, $F = \ln t$, $t > 0$ and*

$$\Gamma t = \begin{cases} \frac{t}{5}, & \text{if } 0 < t \le 0.15 \\ 0.15, & \text{if } 0.15 < t \le 0.17 \\ 0.17, & \text{if } t > 0.17. \end{cases}$$

*Further, $\Phi, \Psi$ satisfy condition $(ii)$ of Theorem 1 with*

$$\kappa(\epsilon) = \begin{cases} 4, & \text{if } \epsilon \ge 0.17 \\ 5 - \epsilon, & \text{if } \epsilon < 1.17. \end{cases}$$

*We also observe that $\lim_{\theta \to 2} \Lambda_0(\theta, 2) \ne 0$ and $\lim_{\xi \to 2} \Lambda_0(2, \xi) \ne 0$.*

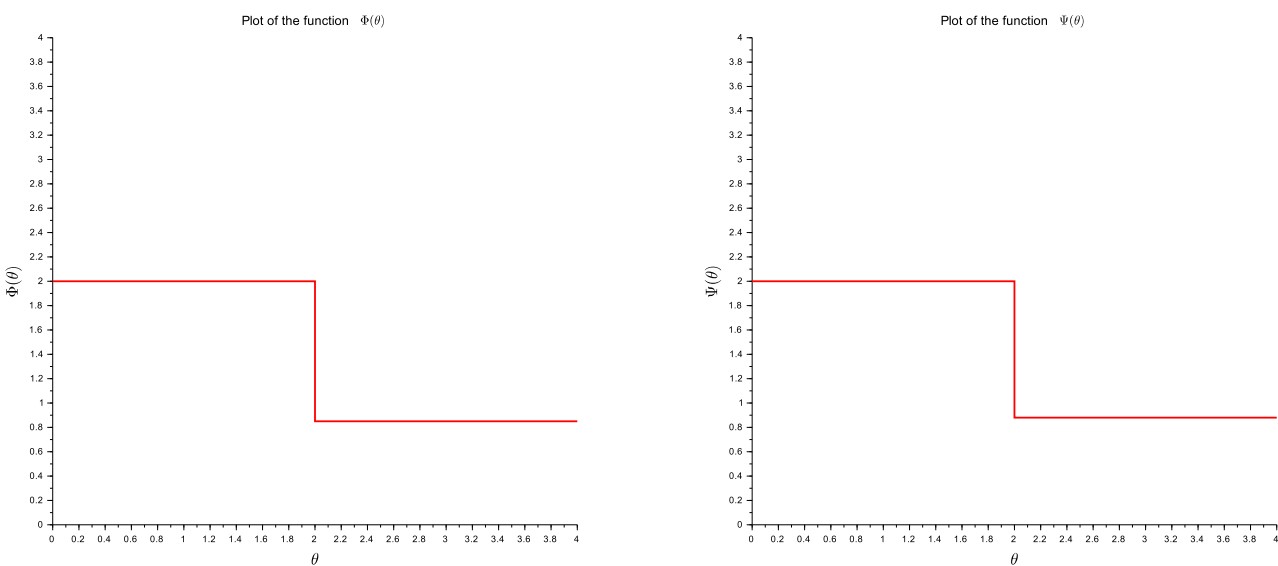

**Figure 1.** Plot of the functions $\Phi$ and $\Phi^2$.

## 3. A Fixed-Circle Result

In situations when the fixed point of a self-map is not unique, the study of geometric properties of fixed points becomes important. In certain cases, the fixed set of a mapping happens to be the unit circle and sometimes the fixed set contains a circle. Such findings initiated the study of fixed-circle problems.

In this section, we present a fixed-circle result by dropping the completeness of the MS. We shall use the following notation which was recently used by Ozgur and Tas [17] to describe some interesting fixed circle-problems, where $\Phi$ is a self-map on the MS.

$$\Delta(\theta, \xi) = \max\{\eta(\theta, \xi), \eta(\theta, \Phi\theta), \eta(\xi, \Phi\xi), \left[\frac{\eta(\theta, \Phi\xi) + \eta(\xi, \Phi\theta)}{1 + \eta(\theta, \Phi\theta) + \eta(\xi, \Phi\xi)}\right]\eta(\theta, \xi)\}.$$

By $C_{\theta_0, \rho}$, we denote a circle $\{\theta \in \mathcal{W} : \eta(\theta, \theta_0) = \rho\}$, whereas by $D_{\theta_0, \rho}$, we denote a disk $\{\theta \in \mathcal{W} : \eta(\theta, \theta_0) \leq \rho\}$.

**Theorem 2.** *Let $(\mathcal{W}, \eta)$ be an MS and $\Phi : \mathcal{W} \to \mathcal{W}$ be a self-map. Define $\rho = \inf\{\frac{\eta(\Phi\theta, \theta)}{\sqrt{2}} : \Phi\theta \neq \theta, \theta \in \mathcal{W}\}$. If there exist $\theta_0 \in \mathcal{W}, \delta > 0$ and $F \in \mathcal{F}$ satisfying*
*(i)     For all $\theta \in C_{\theta_0, \rho}$, there exists $\tau > 0$ such that*

$$\rho \leq \Delta(\theta, \theta_0) < \rho + \tau \implies \eta(\Phi\theta, \theta_0) \leq \rho;$$

*(ii)    For all $\theta \in \mathcal{W}$,*
$$\eta(\Phi\theta, \theta) > 0 \implies \eta(\Phi\theta, \theta) \leq \Gamma(\Delta(\theta, \theta_0)),$$

*where $\Gamma : \mathbb{R}^+ \to \mathbb{R}^+$ has the property $\Gamma(s) < s$ for each $s > 0$.*
*Then, $\Phi\theta_0 = \theta_0$ and $C_{\theta_0, \rho}$ is a fixed circle of $\Phi$. Further, the disk $D_{\theta_0, \rho}$ is fixed by $\Phi$. In addition, $\Phi$ is discontinuous at $\omega \in D_{\theta_0, \rho}$ if and only if $\lim_{\theta \to \omega} \Delta(\theta, \omega) \neq 0$.*

**Proof.** Let $\theta \in C_{\theta_0, \rho}$ and $\Phi\theta \neq \theta$ so that $\eta(\Phi\theta, \theta) > 0$. By condition $(ii)$ of the hypothesis, we have that

$$F(\eta(\Phi\theta, \theta) \leq F(\Gamma(\Delta(\theta, \theta_0)))$$
$$\implies \eta(\Phi\theta, \theta) \leq \Gamma(\Delta(\theta, \theta_0))$$
$$\implies \eta(\Phi\theta, \theta) < [\max\{\eta(\theta, \theta_0), \eta(\theta, \Phi\theta), \eta(\theta_0, \Psi\theta_0), \left[\frac{\eta(\theta, \Psi\theta_0) + \eta(\theta_0, \Phi\theta)}{\eta(\theta, \Phi\theta) + \eta(\theta_0, \Psi\theta_0) + 1}\right]\eta(\theta, \theta_0)\}].$$

Rest of the proof can be obtained in a similar manner as in the proof of Theorem 2.3 in [17]. $\square$

The next example shows that the converse of Theorem 2 is not true in general.

**Example 2.** *Let $\mathcal{W} = \mathbb{R}$ and $\Phi : \mathcal{W} \to \mathcal{W}$ be defined as*

$$\Phi(\theta) = \begin{cases} \theta, & \text{if } \theta \in D_{\theta_0, \rho} \\ \theta_0, & \text{if } \theta \notin D_{\theta_0, \rho}, \end{cases}$$

*where $\rho > 0$.*
*Then, $\Phi$ does not satisfy condition $(ii)$ for any $\kappa > 0$ and $F = \ln t, t > 0$ and $\Gamma(s) = \frac{s}{\sqrt{2}}, s > 0$.*
*However, $\Phi$ fixes every circle $C_{\theta_0, r}$ with $r < \rho$.*

## 4. Conclusions and Future Work

In the current work, we presented some new results on discontinuity at fixed points with the help of *F*-contractive inequalities. Bisht and Pant [25] elucidated actual physical circumstances on the applicability of such discontinuity results. The McCulloch–Pitts model is a widely sought-after and prominent model in Artificial Intelligence and Biology, which describes algorithms for neural networks to reduce and optimize the aberration of neurons from its limiting equilibrium condition. Such a stabilization can be modeled with the help of fixed points of some specific mappings. The functions derived from this procedure exhibit discontinuity at a fixed point the reason of which is a jump in the threshold frequency. Therefore, these discontinuity results always have potential application in neural networks.

We refer to the works listed in [26–28] for details about these models. Obtaining multivalued analogues of the current results using the framework as in [29,30] and the analogues in terms of enriched contractions as in [31] are also interesting suggested future work.

**Funding:** This research received no external funding.

**Institutional Review Board Statement:** Not applicable.

**Informed Consent Statement:** Not applicable.

**Data Availability Statement:** Not applicable.

**Acknowledgments:** The author expresses his hearty gratitude to all the learned referees for their constructive comments which have improved the manuscript. The graphs in this manuscript have been created by the free and open-source software SciLab-6.1.0.

**Conflicts of Interest:** The author declares that he has no known competing financial interests or personal relationships with anyone that could have appeared to influence the work reported in this paper.

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
