# Peer review of "Common Fixed-Point and Fixed-Circle Results for a Class of Discontinuous F-Contractive Mappings"

_mathematics, doi:10.3390/math10091605_

Round 1

Reviewer 1 Report

The  author explored the the fact that a  contractive inequalities those do not imply the continuity of the mapping at the fixed points.  They introduced and  established a common fixed point (CFP) result for a newly introduced  contractive mappings and investigated its discontinuity at the CFP. In addition, they established a fixed-circle result.  The obtained results are interesting and the proofs are correct.

However, F contraction type results have been considered in literature widely and many fixed point results were presented in metric spaces, b-metric and other  spaces. To present a more accurate literature information author(s) should mention the following papers:

1. Some Fixed-Point Results via Mix-Type Contractive Condition, Journal of Function Spaces , 2021.

2.  AFASSINOU K, MEBAWONDU AA, ABASS HA, NARAIN OK. EXISTENCE OF SOLUTION OF DIFFERENTIAL AND RIEMANN-LIOUVILLE EQUATION VIA FIXED POINT APPROACH IN COMPLEX VALUED b-METRIC SPACES.

3. Mebawondu, Akindele Adebayo, Chinedu Izuchukwu, Kazeem Olalekan Aremu, and Oluwatosin Temitope Mewomo. "Some fixed point results for a generalized TAC-Suzuki-Berinde type F-contractions in b-metric spaces." Appl. Math. E-Notes 19 (2019): 629-653.

4. Example 2.3 is more of a constant function. If possible can the author(s) give another interesting example.

5. The authors(s) should take out time to go through the manuscript for typos, as there are lots of them. 

Author Response

Respected reviewer, thank you so much for your constructive report. It has helped me to improved the manuscript a lot. The paper has been revised accordingly. Please check the attached PDF for details of the revision.

Reviewer 2 Report

The work is well and clearly written with seemingly correct evidence. Of course, the author should accept my remarks. It is important to point out that the subject of the research is still quite current because the old question is being discussed:
  whether the given mapping is continuous in its possible fixed point. The same question can be asked in future research in a multifaceted case. I am waiting for a revision of the work and then, if necessary, I can ask for new repairs.

Author Response

(The authors gave the same response as above.)

Reviewer 3 Report

The paper addresses a topic of interest to researchers in this field of research.
The results obtained are worthy of consideration.
The organization of the work should be improved. I believe that the Introduction and the Preliminaries should be two separate sections.
The results seem to be correct, but their presentation must still be made clear and rigorous.
If the figures were obtained with the help of software, then this must be mentioned.
Also, even though I am not a specialist in the use of English, I believe that the expression in English needs to be improved as well.
All this needs to be remedied, especially since the paper is dedicated to a specialist with outstanding results in Fixed Point Theory, Professor B.E. Rhoades.
Only then can we decide on the publication of this paper.

Author Response

(The authors gave the same response as above.)

Round 2

Reviewer 2 Report

As in re-review

Author Response

Respected reviewer, thank you so much for your constructive comments. Please check the attached PDF.

Reviewer 3 Report

The paper can be published.

Author Response

Respected reviewer, thank you so much for your comments and acceptance of the article. Please check the attached PDF as a minor revision has been carried out.
